# Quantum-enabled operation of a microwave-optical interface

Rishabh Sahu [1✉], William Hease[1], Alfredo Rueda[1], Georg Arnold[1], Liu Qiu [1] & Johannes M. Fink [1✉]

Solid-state microwave systems offer strong interactions for fast quantum logic and sensing but photons at telecom wavelength are the ideal choice for high-density low-loss quantum interconnects. A general-purpose interface that can make use of single photon effects requires < 1 input noise quanta, which has remained elusive due to either low efficiency or pump induced heating. Here we demonstrate coherent electro-optic modulation on nanosecond-timescales with only $0.16^{+0.02}_{-0.01}$ microwave input noise photons with a total bidirectional transduction efficiency of 8.7% (or up to 15% with $0.41^{+0.02}_{-0.02}$), as required for near-term heralded quantum network protocols. The use of short and high-power optical pump pulses also enables near-unity cooperativity of the electro-optic interaction leading to an internal pure conversion efficiency of up to 99.5%. Together with the low mode occupancy this provides evidence for electro-optic laser cooling and vacuum amplification as predicted a decade ago.

---

[1] Institute of Science and Technology Austria, am Campus 1, 3400 Klosterneuburg, Austria. ✉email: rsahu@ist.ac.at; jfink@ist.ac.at

uture quantum networks[1] between solid-state quantum systems[2] housed in separated cryogenic environments require efficient, low-noise, and high-bandwidth interfaces to telecom wavelength communication technology[3]. A general-purpose "quantum-enabled" interconnect must add $N_{in} < 1$ noise photons referenced to the input which allows for heralded quantum conversion and teleportation schemes in the presence of practical loss channels[4]. "Quantum limited" conversion with $N_{in} < 0.5$ and close to unity efficiency on the other hand will be required for fully deterministic protocols that also maintain the Wigner negativity of the input state[5,6]. On the practical side, high instantaneous bandwidth is important to be able to shape and match the temporal mode of itinerant microwave quantum states for high-fidelity deterministic quantum network protocols[7].

Growing efforts in a wide variety of systems[8–10] led to record device efficiencies close to 50%[11] using radiation pressure coupling, but the necessary noise properties could not be achieved with the so far reported $N_{in}$ ranging from 3.2[11,12] to ~10[4][13,14]. Recent experiments have used short pump pulses with low duty cycle and cooperativity to reduce the average thermal load[15–17] culminating in a noise level of only 0.57 quanta referenced to a localized piezo-mechanical superconducting qubit excitation[16]. However, this does not circumvent the instantaneous effects such as destroying the qubit state and resonator degradation due to Cooper pair breaking[18], as well as difficult to control detunings of the optical cavity owing to the photo-refractive[19] or thermo-optic effects[20].

In contrast to such tightly integrated on-chip approaches based on superconducting thin films, in this work, the cavity mode volumes are chosen to be large. As a result, the Watt-scale powers of a pulse-shaped optical pump can be used without the aforementioned complications. This allows us to reach unity cooperativity and unlocks the fast and coherent time dynamics of a strongly interacting electro-optic system. Moreover, lowering the microwave mode occupancy for the same cooperativity we observe a transition from the classical to the quantum regime of multi-mode cavity quantum electro-optics[21].

## Results

**Physics and implementation.** Electro-optic converters use the nonlinear properties of a non-centrosymmetric crystal to couple microwave and optical fields[21–23]. We enhance the nonlinearity using a high-quality cavity resonance for the optical and microwave fields[14,18,24–29]. A schematic of the experiment is shown in Fig. 1. The device, which was also used in ref. [14], consists of a lithium niobate whispering gallery mode optical resonator that is clamped by the metallic walls of a superconducting aluminum cavity. The latter is in situ tunable and designed to maximize the mode overlap between microwave and optical modes that respect the phase-matching condition, see Methods for device parameters.

The interaction Hamiltonian is given as $H_{int} = \hbar g_0 (\hat{a}_e \hat{a}_p \hat{a}_o^\dagger + \hat{a}_e^\dagger \hat{a}_p \hat{a}_s^\dagger)$ + h.c., where, $g_0$ is the vacuum coupling rate and $\hat{a}_p$, and $\hat{a}_e$ represents the optical pump mode and microwave mode annihilation operators, respectively. $\hat{a}_o$ (optical signal mode), and $\hat{a}_s$ (Stokes mode) represents the optical mode annihilation operators on the blue and red sides of the optical pump, respectively. The first term in the Hamiltonian is the beam splitter interaction between modes $\hat{a}_e$ and $\hat{a}_o$ with the anti-Stokes scattering rate $\Gamma_{AS}$. The second term corresponds to the two-mode squeezing between modes $\hat{a}_e$ and $\hat{a}_s$ with the Stokes scattering rate $\Gamma_S$. We suppress phase-insensitive amplification owing to $\Gamma_S$ via cross-polarization coupling of the Stokes mode $\hat{a}_s$ with a polarization-orthogonal degenerate optical mode, $\hat{a}_r$[25,30]. This hybridizes the optical mode $\hat{a}_s$ limiting its participation in the dynamics, as schematically shown in the optical spectrum in

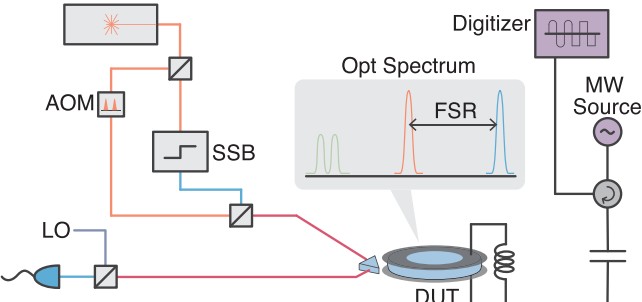

**Fig. 1 Schematic of the experiment.** The laser is divided into two parts: the optical pump and the signal. The pump is pulsed via an acousto-optic modulator (AOM). The signal part is frequency up-shifted using a single sideband modulator (SSB). The optical pump and signal are combined and sent to the device under test (DUT) in a dilution refrigerator via an optical fiber. The microwave signals are sent and received using a single port via a circulator. Optical and microwave output signals are measured with heterodyne detection.

Fig. 1. The Hamiltonian is then given as

$$\hat{H}_{int} = \hbar g_0 (\hat{a}_e \hat{a}_p \hat{a}_o^\dagger + \hat{a}_e^\dagger \hat{a}_p \hat{a}_s^\dagger) + iJ\hat{a}_s^\dagger \hat{a}_r + \text{h.c.}, \quad (1)$$

where $J$ is the cross-polarization coupling rate between the $\hat{a}_s$ and $\hat{a}_r$ modes.

The figure of merit in electro-optic systems is the cooperativity $C = 4\bar{n}_p g_0^2/(\kappa_o \kappa_e)$, where $\bar{n}_p$ stands for the number of optical pump photons inside the optical resonator and $\kappa_o$ ($\kappa_e$) for the total loss rate of the participating optical (microwave) mode. In analogy, we define the cooperativity for characterizing the mode coupling $C_J = 4J^2/(\kappa_s \kappa_r)$. On resonance, the total transduction efficiency including the gain due to amplification is calculated in terms of the two cooperativities as

$$\eta_{tot} = \eta_o \eta_e \Lambda^2 \frac{4C(1 + C_J^{-1})^2}{(1 + C + C_J^{-1})^2}, \quad (2)$$

with the two-mode coupling efficiencies $\eta_i = \kappa_{i,ex}/\kappa_i$ with $i = e, o$ and the factor $\Lambda$ accounting for optical coupling loss due to spatial mode-mismatch of the input beam. The gain due to amplification depends on the suppression ratio $\mathcal{S} = \Gamma_S/\Gamma_{AS} = (1 + C_J)^{-1}$. For $J \to \infty$, i.e., perfect suppression $\mathcal{S} = 0$, Eq. (2) reduces to the simple triply resonant model, as shown in the Supplementary Information, thus defining the (hypothetical) pure conversion efficiency without gain. We furthermore define the internal transduction coefficient by factoring out the measured coupling losses from the measured and calibrated total transduction efficiency, as $\eta_{in} = \eta_{tot}/(\eta_o \eta_e \Lambda^2)$.

**High cooperativity pulsed conversion.** Figure 2a, e shows the calibrated conversion efficiency for microwave-to-optics and optics-to-microwave conversion, respectively, for different cooperativities. The converted pulses are measured for two cases with a 200 MHz bandwidth, i.e., a CW signal (solid lines) and a pulsed signal while the pulsed optical pump pulse is on (dashed lines). The solid and dashed lines show the theoretical prediction from the numerical model with the input optical loss as the only fit parameter. We generally find excellent agreement and assign the observed mismatch in Fig. 2a for the case of pulsed signals to a small amount of uncorrected output filter drift.

We observe a large overshoot at the beginning of the converted pulse for CW signals (solid lines) and a smaller one for high cooperativity pulsed signals (dashed lines). The former appears because the cavity is pre-loaded with photons which get converted immediately when the optical pump pulse arrives. This momentarily

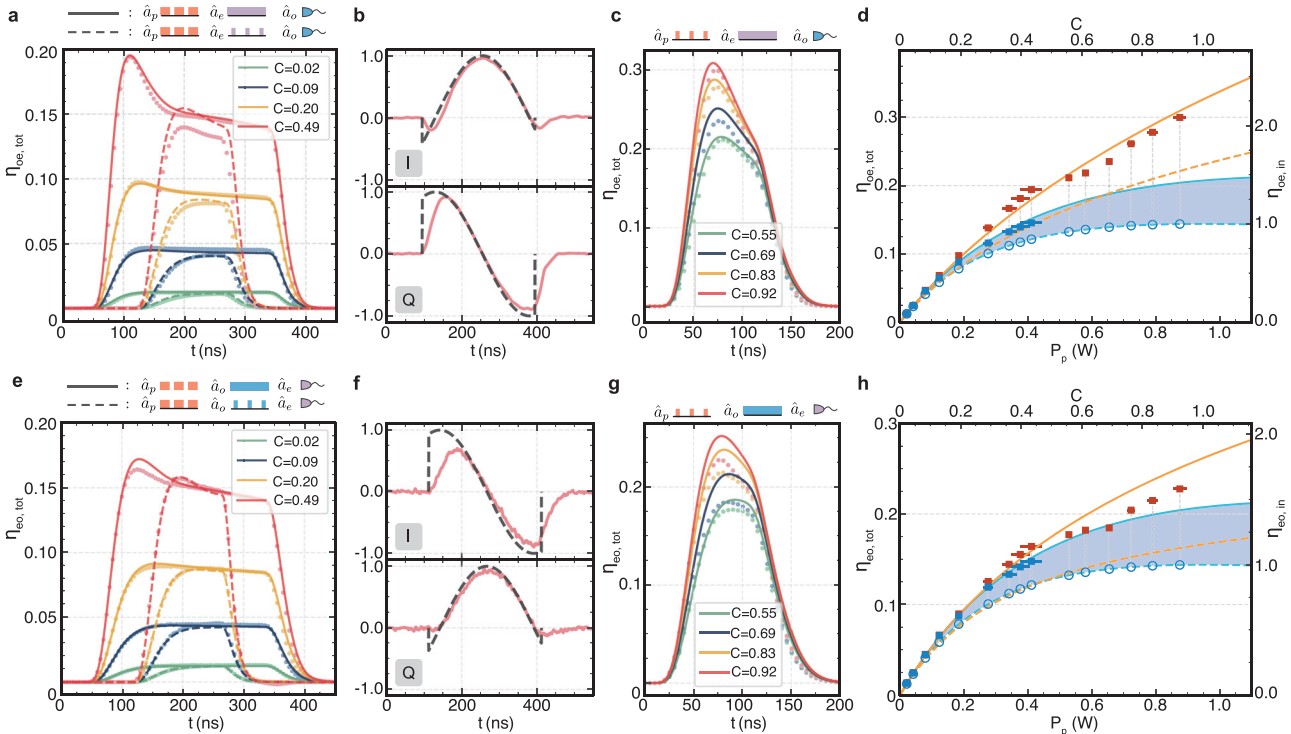

**Fig. 2 High cooperativity bidirectional conversion.** Top (bottom) row panels show results for microwave-to-optics (optics-to-microwave) conversion.
**a, e** Converted signal pulses (bright dots are calibrated measurements and lines are theory) for a 300 ns-long optical pump pulse measured with a 500 Hz repetition rate for a CW signal (solid lines) and a pulsed signal (dashed lines). **b, f** Measured IQ quadrature modulation of the converted signal pulse (red lines) and the applied IQ quadrature modulation to the input signal (gray dashed lines) for $C = 0.49$. **c, g** Converted signal pulses (bright dots are calibrated measurements and lines are theory) for a 100 ns-long optical pump pulse and a CW signal measured with a 500 Hz repetition rate. **d, h** Summary of measured steady-state (blue squares) and peak (red squares) total transduction efficiency (left axis) and internal transduction coefficient (right axis) as a function of the in-pulse pump power $P_p$ and cooperativity. Blue lines show the theoretical prediction for the steady-state (Eq. (2)) and orange lines for the peak values (see Supplementary Information), respectively. Solid lines are theory taking into account the finite suppression ratio $\mathcal{S} = 0.22$, whereas the dashed lines are for the ideal case $\mathcal{S} = 0$. The blue dashed line indicates the achieved $C$-dependent steady-state pure conversion efficiency and the blue-shaded area highlights the calculated gain of the measured steady-state transduction efficiency owing to finite sideband suppression. The vertical (smaller than the symbols) and horizontal error bars represent the standard error of the individual measured time-averaged values.

bypasses the coupling loss to the cavity and since $\eta_o \neq \eta_e$, the magnitude of this overshoot is different for the two directions of conversion. Importantly, there is also an overshoot in the case of pulsed signals (dashed lines) with $C \gtrsim 0.5$ because at high cooperativity the coherent interaction rate approaches that of the two-loss rates. This feature, therefore, represents the onset of coherent oscillations between microwave and optical photons in a strongly coupled electro-optic system.

For the measurement with $C = 0.49$, which corresponds to a parametrically enhanced coupling strength of $\sqrt{\bar{n}_p}g_0/(2\pi) = 6.58$ MHz with $\bar{n}_p = 3.16 \times 10^{10}$ pump photons for a pulse pump power of $P_p \approx 0.4$ W, we explicitly show real-time complex quadrature control in Fig. 2b, f, as required for high-fidelity quantum communication protocols. A linear phase change is imprinted on the input signal pulses while keeping the amplitude constant. The measurements of the two converted quadratures match the input modulation closely. The only exception is during the beginning and the end of the optical pump pulse owing to the finite transducer bandwidth of 18 MHz for this $C$.

In order to reach $C \approx 1$, without entering an observed regime of instability, we use shorter, i.e., 100 ns-long optical pump pulses as shown in Fig. 2c, g. The highest microwave-to-optical conversion efficiency momentarily reaches up to 30% for a CW signal tone. In very good agreement with theory, this is a result of three effects, i.e., the pre-loading of the microwave cavity, the increased gain for higher $C$, and the strong and coherent electro-optic interaction, which would

reveal an oscillatory behavior if longer pulses could be sustained. The observed deviations from theory in Fig. 2g are caused by the slight broadening of the microwave mode linewidth owing to the increased average bath temperature, in agreement with CW pump experiments[14], an effect which the theory model does not take into account. The Supplementary Information shows measurements up to $C = 1.2$ where the optical Kerr effect leads to instability and amplification for this particular pump pulse length.

A summary of the measured transduction efficiencies as a function of the applied optical pulse power $P_p$ and corresponding cooperativity $C$ is presented in Fig. 2d, h. The red and blue colors show measured peak and steady-state values, respectively. Solid lines are predicted transduction efficiencies for our experimental parameters corresponding to $\mathcal{S} = 0.22$, whereas dashed lines represent the case of perfect suppression $\mathcal{S} = 0$. The blue-shaded area thus represents the gain owing to the not fully suppressed Stokes process. For $C \gtrsim 0.35$, the achieved internal transduction coefficient $\eta_{in}$, which we define operationally by dividing the measured $\eta_{tot}$ by the measured coupling losses, can exceed 1. This is owing to signal pre-loading into the cavity and coherent electro-optic oscillations in case of the extracted peak values (red symbols), as well as due to gain. This is explained in detail in the Supplementary Information.

The blue circles indicate the corresponding pure conversion efficiencies (without gain) for itinerant photons up to $C = 0.92$, reaching $\eta_{in} = 0.995$ and $\eta_{tot} = 0.144$. As pure conversion

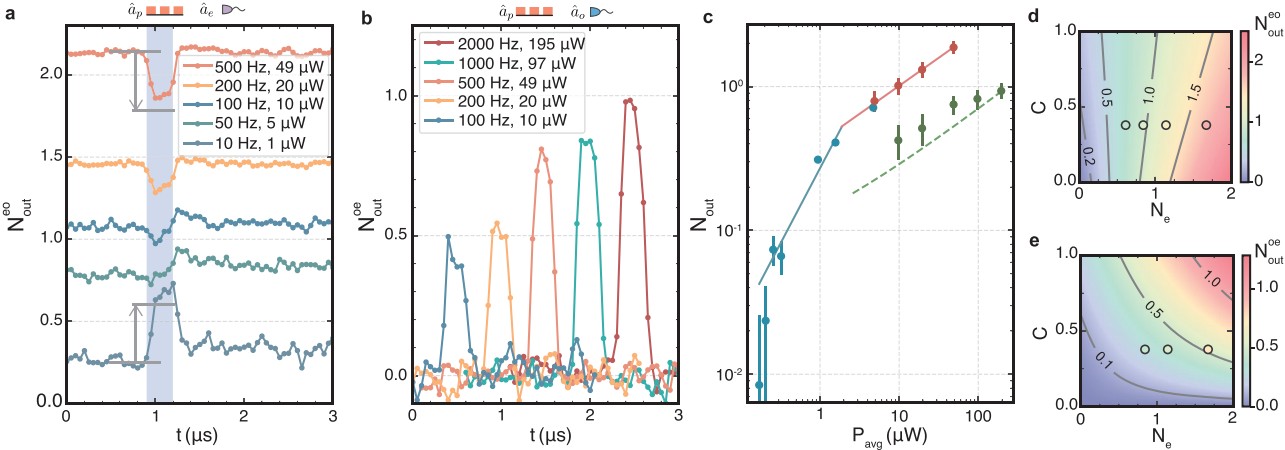

**Fig. 3 Measured added noise during conversion. a** $N_{out}^e$ for a 300 ns-long optical pump pulse ($C = 0.38$) for different repetition rates $f_{rep}$ ($P_{avg}$) measured with a 10 MHz bandwidth centered at the microwave resonance. The blue-shaded region shows the time span when the optical pump is on. The gray arrows mark the noise change as predicted from theory. **b** $N_{out}^{oe}$ for the same optical pump pulses and various $f_{rep}$ ($P_{avg}$) measured with a 10 MHz bandwidth centered at the optical signal frequency. The noise pulses are separated in time for better visibility. **c** Compilation of all measured $N_{out}$ for different time-averaged optical pump powers $P_{avg}$. Red and blue points represent measured $N_{out}^e$ with 10 MHz and 100 kHz bandwidths, respectively. Green points represent $N_{out}^o$ measured with 10 MHz bandwidth. The solid lines represent power-law fits. The green dashed line shows the predicted $N_{out}^o$ using the measured $N_{out}^e$. All the points are taken with $C = 0.38$ ($\eta_{tot} = 11.4\%$) except for the first three blue points which correspond to $C = 0.20$ ($\eta_{tot} = 8.0\%$), 0.24 ($\eta_{tot} = 9.1\%$), and 0.3 ($\eta_{tot} = 10.3\%$), respectively. The error bars are inferred as in ref. [14]. For low occupancies, the confidence interval is given by the systematic error due to the absolute drift of the measurement baseline of $\pm 0.02$ photons. **d** Density plot of the calculated $N_{out}^e$ (10 MHz bandwidth) as a function of $C$ and $N_e$ for $S = 0.22$. The top four measurements of $N_{out}^e$ for the same $S$ taken from **a** are shown as circles with experimental values represented by the inside color. **e** Density plot of the calculated $N_{out}^o$ (10 MHz bandwidth) as a function of $C$ and $N_e$ for $S = 0.22$. The left-most three measurements from **b** are shown as circles where experimental values are represented by the inside color.

efficiency estimates are based on a simplified theory calculation, we decided to instead focus on the measured steady-state transduction values and take into account the finite gain when calculating the corresponding added conversion noise, as explained below. The measured higher efficiencies due to signal pre-loading on the other hand are relevant to be able to make a fair comparison with the performance of current[16] and future interfaces that produce the signal photons inside the cavity and thus circumvent the microwave coupling losses.

**Thermal and quantum noise**. Figure 3 captures the effect of the two relevant types of noise. The first is thermal noise due to optical pump absorption heating, which depends on the time-averaged optical power $P_{avg} = P_p t_p f_{rep}$ with the pulse length $t_p$ and the repetition rate $f_{rep}$. The second contribution is amplified vacuum noise due to finite transducer gain, which depends on $S$ and $P_p$. The shown output noise photon numbers are defined by integrating the noise emission spectrum $S_{ii,out}$ (see Supplementary Information) over the Gaussian measurement filter function $\chi$ as $N_{out}^i = \int_{-\infty}^{+\infty} \chi(\omega) S_{ii,out}(\omega) d\omega$.

Figure 3a shows the microwave noise output $N_{out}^e$ in the time domain, measured with a 10 MHz filter bandwidth (100 ns time resolution) centered at the microwave resonance, when 300 ns-long optical pump pulses with $C = 0.38$ are applied (shaded region) with different repetition rates from 10 to 500 Hz. High repetition rates increase the measured average thermal output noise, which stays approximately constant during the measurement time of 3 µs. During the pulse, however, we observe either a classical or a quantum effect depending on the average thermal occupancy of the microwave mode. For higher mode temperature (red curve), parametric laser cooling of the microwave mode[21], owing to upconversion of noise to optics is dominant and in agreement with theory (gray arrow). But as the thermal noise is decreased for the same cooperativity, additional noise due to vacuum amplification overwhelms the parametric cooling effect. For the lowest occupancy curve with a 10 Hz repetition

rate, the vacuum amplification is clearly observed during the pump pulse and is in good agreement with theory (gray arrow). This last curve is measured with a lower suppression $S \approx 0.82$ (using a different set of optical modes with a different magnitude of avoided crossing) in order to enhance the effect for a better signal-to-noise ratio. We assign slight mismatches between the theoretical prediction and experiment to an expected small amount of thermal heating during the pulse.

The observation of parametric laser cooling implies the presence of noise at the output of the optical signal mode $\hat{a}_o$. $N_{out}^o$ is measured with a 10 MHz bandwidth around the optical signal frequency and shown in Fig. 3b. The optical pump pulses are the same as in Fig. 3a with varying repetition rates. With higher $P_{avg}$, the thermal microwave mode occupancy increases, thus, increasing the output optical noise.

We summarize these results in Fig. 3c as a function of $P_{avg}$. The $N_{out}^i$ during the pulse from Fig. 3a are shown as red points, and from Fig. 3b as green points, along with corresponding power-law fits (solid lines). The dashed green line represents the predicted optical noise based on theory and the fitted microwave noise. In addition, we show microwave noise measurements conducted with a 100 kHz bandwidth for a better signal-to-noise ratio at the lowest occupancies where the system is deep in its quantum ground state (blue points), in agreement with ref. [14]. Unlike the triggered 10 MHz measurements, the 100 kHz measurements run continuously to sample the maximum of the emission noise spectral density due to the average thermal noise from the triggered optical pump pulses, resulting in (depending on $C$) up to ≈ 20% higher noise photon number values compared to the high-bandwidth measurements and thus represent a worst-case scenario for bandwidth matched time-domain conversion experiments. We report these higher values for the lowest achieved added conversion noise photon numbers.

In Fig. 3d, e, we show the theoretical prediction of the $N_{out}^e$ and $N_{out}^o$, respectively, as a function of the steady-state microwave mode occupancy $N_e$ and cooperativity $C$ for the relevant

experimental case $\mathcal{S} = 0.22$. As a function of $N_e$, in Fig. 3d the contours change from left-leaning to right-leaning as the transition from quantum amplification to classical cooling occurs. The measurements from Fig. 3a are shown as circles where the inside color represents the experimentally measured value in excellent agreement with the theoretical prediction. Similarly, the predicted dependence of $N^o_{out}$ is shown in Fig. 3e where we included the measurements from Fig. 3b with good agreement with theory. Low bandwidth measurements (blue points in Fig. 3c) do not contain time-domain information and were therefore not included. These results provide strong evidence that $P_{avg}$ fully determines the thermal noise limitations of this device. We furthermore find that the dilution refrigerator baseplate temperature, which went up to at most 60 mK for the highest $P_{avg}$, follows the same curve as reported in ref. [14] for continuous optical pump fields. This observed agreement with $P_{avg}$ is expected due to the short pump timescales and the relatively high heat capacity leading to slow thermalization. The large difference between mode temperature $N_e$ and the relatively cold dilution refrigerator is also owing to finite thermalization between the dielectric cavity where the heat originates, the superconducting cavity walls, and the cold dilution refrigerator baseplate bath.

Low occupancies and high cooperativity are the preconditions for interesting cavity quantum electro-optics experiments in analogy to cavity optomechanics[21] as well as for quantum-limited conversion. Figure 4a shows the calculated quantum cooperativity $C_q = C/N_e$—a measure of the electro-optic state transfer rate compared to the thermal decoherence rate of the microwave mode[31]—for the measurements in Fig. 3c, where $N_e$ is inferred from the measured $N^e_{out}$. The achieved large $C_q \gg 1$ are a result of the low mode occupancies and encourage further investigations in the direction of two-mode squeezing of hybrid microwave and optical field states[32].

The most relevant quantity that signifies quantum-enabled conversion is the resulting equivalent added noise photon number referenced to the input of the converter where a non-classical input signal would be applied $S_{out}(\omega) = \eta_{tot}(S_{in}(\omega) + N_{in})$. The two measured $N^i_{out}$ include both thermal and quantum noise contributions and we calculate the resulting equivalent added input noise photon number with $N^{ij}_{in} = N^i_{out}/\eta_{tot}$. Low values of $N^{ij}_{in}$ therefore require simultaneously high efficiency and low output noise, which is achieved in this work.

Figure 4b, d show $N^{ij}_{in}$ for optics-to-microwave ($N^{eo}_{in}$) and microwave-to-optics ($N^{oe}_{in}$) conversion respectively as a function of $N_e$ and $C$ for $\mathcal{S} = 0.22$. The lowest five blue points from Fig. 3c are marked with crosses as the parameters we achieved experimentally. For $N^{eo}_{in}$, we reach $1.11^{+0.15}_{-0.07}$, and for $N^{oe}_{in}$, we reach as low as $0.16^{+0.02}_{-0.01}$ equivalent added noise photons. Here, the confidence interval is taken from error propagation using the confidence interval of the measured $N^i_{out}$.

## Discussion

We have demonstrated a modular electro-optic transducer that converts itinerant photons from the microwave X to the telecom C band (and reverse) with a total bidirectional conversion efficiency of $\eta_{tot} \approx 15\%$ (up to 30% for pre-loaded cavity states) with the demonstrated equivalent input noise $N^{oe}_{in} < 1$ and a quantum cooperativity $C_q \gg 1$ for a cooperativity $C = 0.38$. Together with the phase-coherent control of the spectral and temporal dynamics with a bandwidth of up to 24 MHz, it is suitable for optically heralded superconducting quantum networking schemes[33] and the generation of non-classical correlations.

The new parameter regime of near-strong coupling cavity quantum electro-optics also enables the first systematic studies of dynamical and quantum back-action in electro-optic systems[21] as well as the deterministic generation of microwave-optics entanglement in the continuous variable domain[32]. This might lead the way for the scaling-up of error-protected superconducting quantum processor modules in large data centers but it could also find other applications such as for quantum-enhanced remote sensing[34] or sensitive radiometry[35], as well as for fiber-based classical control and readout of cryogenic qubit processors[36–38].

Current limitations include the somewhat complex assembly, the need for high pump powers, which leads to relatively slow pulse repetition rates, and the finite suppression ratio of the Stokes process, which is the dominant source of added noise at low average pump power. In future, these challenges can be addressed with more robust and compact packaging such as the proposed center-clamped design[32] that should also reduce phonon losses that might currently limit the microwave mode quality factor[39]. Such a thin-film approach would furthermore facilitate the use of thinner optical resonators with a larger electro-optic coupling that can help to avoid the optical instability owing to the Kerr effect.

Fabrication and material improvements have already led to optical quality factors that are 40 times higher than reported here. Maintaining this in a full device at low temperature would reduce the required pump power by three orders of magnitude for the same $C$, thus significantly reducing the thermal load and potentially allowing for much faster repetition rates. Narrower optical linewidths could also help to further improve the suppression of the Stokes process and the resulting amplified vacuum noise. Higher internal quality factors will furthermore be key to achieving higher external mode coupling ratios $\eta_{o,e}$ with the same bandwidth—a necessary requirement to push the total conversion efficiency for itinerant photons towards unity. This is needed for fully deterministic quantum interconnects operating in the quantum limit.

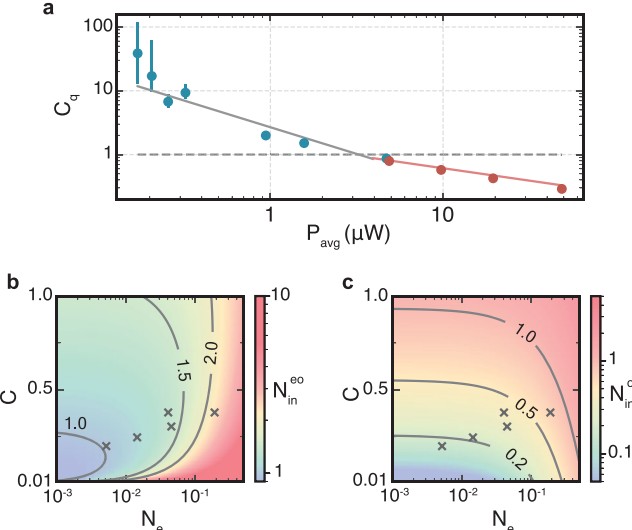

**Fig. 4 Quantum cooperativity and equivalent input added noise. a** Inferred quantum cooperativity $C_q = C/N_e$ for the measured output noise values shown in Fig. 3c. **b, c** Inferred optics-to-microwave $N^{eo}_{in}$ (microwave-to-optics $N^{oe}_{in}$) equivalent added input noise as a function of $C$ and $N_e$ for $\mathcal{S} = 0.22$. Crosses mark the lowest five blue measurement points in Fig. 3c.

## Methods

In this study, we use a device similar to ref. [14]. The optical WGMR is made from z-cut lithium niobate and enclosed in a machined aluminum cavity, which is designed to maximize the field overlaps and match the optical free spectral range

(FSR). The microwave mode is frequency tunable with internal linewidth $\kappa_{e,in}/(2\pi) = 8.1$ MHz, coupling efficiency $\eta_e = 0.41$ and mode frequency $\omega_e/(2\pi) = 8.795$ GHz that accurately matches the optical FSR. The optical WGMR supports high-quality modes with an internal linewidth of $\kappa_{o,in}/(2\pi) = 10.8$ MHz. The optical input and output are coupled to the WGMR mode via frustrated total internal reflection through an in situ movable diamond prism with $\Lambda \approx 0.78$ and $\eta_o = 0.58$ in this experiment. The optical mode $\hat{a}_r$ has a total linewidth of $\kappa_r/(2\pi) = 11.7$ MHz and is coupled to mode $\hat{a}_s$ with a coupling rate of $J/(2\pi) = 27$ MHz. These values are obtained from a comprehensive set of characterization measurements, which include phase-coherent time-domain 2 port scattering parameter measurements at $C \approx 1$ with an excellent agreement to theory, as reported in the Supplementary Information.

## Data availability

All data sets and analysis files used in this study are available at https://doi.org/10.5281/zenodo.5984859.

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

## Acknowledgements

The authors thank S. Wald and F. Diorico for their help with optical filtering, O. Hosten and M. Aspelmeyer for equipment, H.G.L. Schwefel for materials and discussions, L. Drmic and P. Zielinski for software support, and the MIBA workshop at IST Austria for machining the microwave cavity. This work was supported by the European Research Council under grant agreement no. 758053 (ERC StG QUNNECT) and the European Union's Horizon 2020 research and innovation program under grant agreement no. 899354 (FETopen SuperQuLAN). W.H. is the recipient of an ISTplus postdoctoral fellowship with funding from the European Union's Horizon 2020 research and innovation program under the Marie Skłodowska-Curie grant agreement no. 754411. G.A. is the recipient of a DOC fellowship of the Austrian Academy of Sciences at IST Austria. J.M.F. acknowledges support from the Austrian Science Fund (FWF) through BeyondC (F7105) and the European Union's Horizon 2020 research and innovation programs under grant agreement no. 862644 (FETopen QUARTET).

## Author contributions

R.S., W.H., and A.R. worked on the setup and performed the measurements. R.S. did the data analysis. R.S., A.R., and L.Q. developed the theory. G.A. helped with the heterodyne calibration. R.S. and J.M.F. wrote the manuscript with contributions from all authors. J.M.F. supervised the project.

## Competing interests

The authors declare no competing interests.
