## [Peer Review File · Nature Communications]

Reviewers' Comments:

Reviewer #1:

Remarks to the Author:

The authors describe the operation of an electro-optic converter between microwave and optical frequencies. They characterize the conversion efficiency and added noise of the system for various pump power and cooperativities, including up to the regime of $C \sim 1$. This manuscript is a nice follow-up of their previous work (ref 15) and gives a more complete picture of the capabilities of this system, including in the pulsed regime. The experimental techniques are sound and the results are carefully analyzed and compared to theory. However, some parts of the manuscript can be made clearer, and I also have some concerns about how the significance of the results are presented:

1. I found the summary in the abstract and conclusions somewhat misleading. For example, in the second to last paragraph, a total efficiency of 15%, microwave to optical input noise of $\ll 0.5$, and $C_q \gg 1$ are claimed. However, these are not achieved for the same set of parameters. From Fig. 2, the 15% efficiency is achieved at pump powers of ~ 0.4 W, while Fig. 4, $C_q \gg 1$ and low added noise is only achieved for powers $\ll 1$ uW. The abstract similarly does not make this distinction clear. To achieve the quoted input noise of 0.16 quanta, the corresponding conversion efficiency is much less than unity. Furthermore, it does not specify that the input noise quoted is only for microwave to optical conversion.
2. The description of the systems is quite brief, which is ok since it's similar to Ref. 15. However, it's unclear from the manuscript if there were some improvements from the previous publication that enabled this work, and if so, what they were. From what I can tell, the cooperativities achieved in Ref 15 were lower, but I cannot figure out what exactly changed. It would be very helpful if the authors could say a few words about this, perhaps in the methods section.
3. Fig. 3 presents the noise measured over a 10 MHz or 10 kHz bandwidth. It would be good to see a PSD of this noise so that we know what this bandwidth actually includes. According to Supplementary Table 2, the microwave mode linewidth is 13.7 MHz, so I'm not sure how these measurement bandwidths are chosen, and why the noise quanta measured with these two very different bandwidths can be directly compared, since neither is much larger than the linewidth.
4. Can the authors say a bit more about what the significance or usefulness of the peak conversion efficiencies are? It doesn't seem like "pre-loading" the cavity with earlier photons should count toward the conversion efficiency of deterministically converting a particular signal photon. Unless there is a situation in which this is useful, I feel like the emphasis put on this in Fig 2c, d, g, and h might be misleading.
5. I found the discussion of Fig 4b and c somewhat confusing. There's a sentence that says "Use N_e and C , we can also infer the N_{out} that includes both thermal and quantum noise contributions..." Is this inferred N_{out} and the measured N_{out}^e the same, but just inferred for more values of N_e and C than what was measured?
6. What is the meaning of the time delay between the peaks for different powers in Fig 3.b?
7. In the Supplement, the authors mention that the signal and LO are copropagating for the case of low cooperativity to maintain phase stability. Is this not a concern for higher cooperativities?

Some other small things:

1. I also could not find the meaning of the grey horizontal dashed line in Fig. 3.
2. What is P_p in the last paragraph of page 3?
3. "grin" should be an acronym in the caption to Supplementary Figure 2.

Reviewer #2:

Remarks to the Author:

The manuscript presents the coherent transfer of weak coherent radiation through the electro-optical interface based on non-centrosymmetric crystals inside the dilution refrigerator that is promising for the direct interconnection of optical and microwave platforms. The result represents an original and important step in this direction at a nanosecond time scale and with a mean number of input quanta much less than unity. Optical and microwave heterodyne detection brings sufficient information to evaluate this milestone experiment regarding efficiency and added noise.

The manuscript is well written, contains all relevant experimental details, and overcomes noisier, although more efficient, alternatives. As such, it is a timely, relevant and stimulating experimental result. It is worth being published in Nature Communication after solving these comments:

1) It will be better for a general reader to define the main text's visibly added noises N_{in} and N_{out} by formulas as they are central figures of merit.

2) It will be stimulating for the community and also very good for the conclusion of the manuscript if critical predictions for the future transfer of at least nonclassical auto-correlation functions $g^{(2)}$ of single optical (microwave) photon to microwave (optical) output are estimated. The cross-correlation function after the conversion can also be evaluated for comparison. Also, will such an interface already allow transduction of Gaussian or single-photon entanglement?

3) Based on the previous point, it will be good to specify then which regime of the device is optimal for particular tasks in 2).

4) Finally, it will be good to comment on a technical prospect of this device shortly. Can the noise be further lowered, and how? Can the total transduction efficiency be further increased, and how? Can we expect that the interface will improve in both parameters simultaneously soon?

After solving all that points, the manuscript is suitable for Nature Communication.

Reviewer #3:

Remarks to the Author:

The manuscript by Sahu, et al., presents an electro-optic system for quantum conversion between GHz microwave signals and telecom optical photons. The converter system harnesses a 3D cavity design that couples a bulk lithium niobate (LiNbO₃) resonator with a superconducting aluminum cavity via the electro-optic effect. Although the converter is based on the author's previous design published in PRX Quantum 1, 020315 (2020), the authors have done substantial improvement here to push the device performance to close to the quantum-enabling regime. In particular, the authors implemented a pulsed optical pump scheme to boost the pump power, and hence the electro-optic cooperativity to near unity, without introducing a large amount of extra heat or thermal noise. As a result, a high conversion efficiency around 15% with impressively low subphoton added noise (referenced to the input).

The study of microwave-optical quantum conversion has recently become a pivotal research field that attracted a lot of attention. The realization of such quantum converters would enable powerful technologies, such as the combination of the state-of-the-art superconducting quantum computing units and long-distance optical quantum communication channels for building a future quantum network. This, nevertheless, is a very challenging task because of the demanding requirement of a high efficiency (at least 50%) with low added noise smaller than one for quantum signal conversion. So far, a quantum converter has not been demonstrated yet.

This work, in my opinion, represents a milestone effort toward the realization of microwave-optical quantum conversion. To my knowledge, this is the first demonstration of subphoton input added noise among all types of bidirectional microwave-optical converters. At the same time, a decent high conversion efficiency $\sim 20\%$ (although still below the 50% threshold) was achieved. These are very exciting and encouraging results that show the potential and advantages of the bulk electro-optic device design. Therefore, I strongly support its publication in Nature Communications.

The manuscript is well-written, and the data and results presented are solid and convincing. Below are some comments and questions that I think should be addressed before its publication.

1. I feel it inaccurate to claim "Quantum-enabled operation" in the title. As far as I understand, even with zero added noise, a linear converter will need to have at least 50% efficiency in order to convert quantum signals (in other words, a non-zero quantum channel capacity). The efficiency and added noise achieved in this work are obviously below that threshold, which means the device

here won't be able to convert quantum information. Also, I don't think I can consider any of the characterizations and measurements presented in the manuscript as quantum operations unless I missed something quantum-enabled.

2. In the abstract, the authors claimed that "We achieve unity internal photon conversion efficiency by pushing the resonantly enhanced electro-optic interaction all the way to unity cooperativity". A few comments here. (a) First, in the main text, it seems the highest cooperativity achieved is 0.92 – it is a bit rough to simply approximate 0.92 to one. Although extra data for $C = 1.21$ was shown at the end of the Supplementary Information, the system became unstable for conversion operation due to the optical parametric amplification. So, there is actually no data to support that the cooperativity can be continuously increased from 0.92 across 1 while maintaining the system in a stable condition. (b) I don't see any definition or discussion about the "internal efficiency" through the manuscript. Although in a conventional EO converter the internal efficiency can be simply defined as $4C/(1+C)^2$, it is not straightforward from Eq. (2) in this work since the an extra coupling mode is involved with cooperativity C_J and gain. How do the authors define "internal efficiency" in presence of the amplification from the Stokes mode? And will such definition still be useful if the "internal efficiency" may be larger than one? If authors don't want to discuss the internal efficiency, "unity internal photon conversion efficiency" should not be stated in the abstract. Instead, highlighting the total efficiency could be more useful. (c) If I understand, the lowest input added noise of 0.16 was not obtained in the same condition as the highest conversion efficiency – the lowest noise was achieved at $C=0.2$, which gives a lower efficiency of 8%, while the highest efficiency (30%) was when $C=0.92$. So, it is a bit misleading in the abstract to emphasize the low added noise ("...with only 0.16 input noise quanta...") then immediately state that a high "unity internal photon conversion efficiency" was also achieved. The author should indicate that the highest efficiency was obtained with a larger pump power.

3. It is amazing that the authors were able to send Watt-scale optical pump powers to the dilution fridge (even in the pulsed mode) while keeping the fridge at \sim mK. The authors only provided the averaged pump power in the figures, but little information about how the pump affects the fridge and device was given. Although the average power could be as low as a few micro-Watt, there could be certain amount of light absorbed/scattered at the device interface, etc. Even 1% of one Watt (namely, 10mW) is significant for a dilution fridge because the typical cooling power at 20mK is only about 20uW. Can the authors comment on this? I hence believe that there must be non-negligible temperature increase of the base plate and the device at least for the high repetition rate measurement. Was there a thermometer directly mounted on the Al cavity for the authors to monitor its temperature? Or can the authors infer the temperature of the device based on the measured thermal noise N_e ? From Fig. 3(d,e), it seems the largest N_e is about 1.7 (I actually have a question about this, see next comment), corresponding to \sim 1K for a 8.8GHz mode, which is quite significant increase of temperature. I suggest the authors to explain these aspects and add relevant discussions in the main text, which will be useful to readers.

4. I am confused about the N_e label in Fig. 3(d,e) and Fig. 4(b,c). In Fig. 3(d,e), the N_e (circles in the plots) look much larger than the crosses in Fig. 4(b,c). But the figure captions said that they represent the same data in Fig. 3(c). So, was the x-axis label in Fig. 3(d,e) wrong? Another confusing part is that it is not clear which points in Fig. 3(c) correspond to the four circles in Fig. 3(d) since there are 8 points (four blue four red) in Fig. 3(c) for $C=0.38$. Also, in Fig. 4(b,c), do the five crosses correspond to the first five blue dots in Fig. 3(c) (there are seven blue dots in total)?

5. Although the device details are shown in the Methods, it would be convenient to give one- or two-sentence description of the device in the main text for readers who are not familiar with the authors' previous work.

6. In Fig. 2, the color contrast for the measurement and theory lines are a little too low to be seen especially for the light/dark orange lines. Another thing is that the caption says "bright dots are calibrated measurements" – I actually couldn't find any "bright dots" at the beginning, then I realized that the dots are so dense that they form a continuous thick line with light color. Please somehow make them clearer.

7. In Fig. 2(d,h), could the authors explain why the dashed orange lines can be larger than one for the internal efficiency? Is it because of the preloaded cavity photons? However, it barely makes sense to me to define an internal efficiency that can be larger than one. The author may want to re-define it and scale it down to within one so that it can be used to compare with other converters.

8. On Page 3, bottom left, it would be useful to give the value of the absolute pump power used for 3.16×10^{10} pump photons. I guess it would be around one Watt?

9. I saw that different suppression ratio S was used throughout the experiments. I am wondering how the authors control or tune S ? I thought it is determined by the cross-polarization coupling rate J , which is fixed after the lithium niobate resonator is fabricated.

10. In the conclusion paragraph, I am not sure that the authors can state the input added noise is "much smaller" than 0.5 ($0.16 < 0.5$). It might be okay to write " $N^{\text{oe}} < 0.5$ " or " $N^{\text{oe}} \ll 1$ ".

11. In the conclusion paragraph, can the total efficiency of 15% and lowest input noise N^{oe} be obtained at same condition (namely, with same C and S)? I thought that the 15% efficiency is when $C=0.49$, and the 0.16 added noise is when $C=0.2$ which only gives 8% efficiency. Please clarify this to avoid any misunderstanding.

12. As I understand, the microwave cavity is made from Al, which should be superconducting below 1.2K. However, the intrinsic microwave Q factor seems very low, only ~ 1000 . So, is the Al cavity superconducting during the experiments? If yes, then what are the major microwave loss sources? From the bulk lithium niobate?

13. At the end of the main text, could the authors provide some discussions regarding how to further improve the device performance to achieve a real quantum converter? For example, there seems to have quite some room to improve the microwave Q , which will help boost the cooperativity and also lower the pump power needed. I think such discussions could benefit the researchers in community to move forward.

14. In the SI experimental setup figures, could the authors provide more information about the filters $F1$, $F2$? What are the model and specifications (bandwidth, suppression ratio, etc)?

15. In the SI Note 7 Kerr Effect, the authors state "For longer optical pump pulses, the cooperativity threshold for amplification becomes smaller and well below unity." Could the authors explain more about this? Does this mean for longer pump pulses, $C=1$ can no longer be achieved?

Reviewer 1

The authors describe the operation of an electro-optic converter between microwave and optical frequencies. They characterize the conversion efficiency and added noise of the system for various pump power and cooperativities, including up to the regime of $C \sim 1$. This manuscript is a nice follow-up of their previous work (ref 15) and gives a more complete picture of the capabilities of this system, including in the pulsed regime. The experimental techniques are sound and the results are carefully analyzed and compared to theory. However, some parts of the manuscript can be made clearer, and I also have some concerns about how the significance of the results are presented:

We thank the reviewer for these encouraging comments. Below, we have tried to answer all the concerns raised by the reviewer.

Q1: I found the summary in the abstract and conclusions somewhat misleading. For example, in the second to last paragraph, a total efficiency of 15%, microwave to optical input noise of $\ll 0.5$, and $C_q \gg 1$ are claimed. However, these are not achieved for the same set of parameters. From Fig. 2, the 15% efficiency is achieved at pump powers of ~ 0.4 W, while Fig. 4, $C_q \gg 1$ and low added noise is only achieved for powers $\ll 1$ uW. The abstract similarly does not make this distinction clear. To achieve the quoted input noise of 0.16 quanta, the corresponding conversion efficiency is much less than unity. Furthermore, it does not specify that the input noise quoted is only for microwave to optical conversion.

A1: We thank the reviewer for bringing up a possible point of confusion. In Fig. 2, we show the power of the optical pump during the pulse while in Fig. 4, we show the time averaged power during the measurements, i.e. $P_{avg} = P_{pulse} * pulse_length / trigger_time$, which takes into account the waiting time between two subsequent experiments. Working with low duty cycles, the average powers can be extremely small even though the peak pulse powers are large, which warrants simultaneously high efficiency and low added noise at the expense of a reduced duty cycle.

=> In the abstract, we now report both the added noise and the conversion efficiency valid for the same configuration in the microwave to optics direction.

=> In the main part of the manuscript, we have made the definition and distinction between peak and average power more clear using the labels: P_p and P_{avg} and we now provide the formula relating them to the pulse length and repetition rate in the subsection "thermal and quantum noise".

Q2: The description of the systems is quite brief, which is ok since it's similar to Ref. 15. However, it's unclear from the manuscript if there were some improvements from the previous publication that enabled this work, and if so, what they were. From what I can tell, the cooperativities achieved in Ref 15 were lower, but I cannot figure out what exactly changed. It would be very helpful if the authors could say a few words about this, perhaps in the methods section.

A2: In Ref. 14 (previously Ref. 15), we worked with a continuous wave optical pump in the low cooperativity limit. In the new work we use a pulsed pump of higher power and achieve three orders of magnitude higher cooperativity and efficiency using the same physical device. Slight differences include modified coupling configurations (adjusted with piezo positioners) and the use of a different mode splitting (i.e. we work with a slightly different absolute optical frequency). The major improvements are

- i) the frequency locked optical setup that allows to apply Watt scale time domain shaped pulses with nanosecond resolution, as well as
- ii) very high resolution bandwidth detection with comparably low noise, together with
- iii) a multi-mode theory model that enables us to back out the noises correctly (even at high C) and a full time-dependent theory model that also predicts frequency shifts to fully understand the coherent dynamics of the multi-mode system.

=> We now make the changes and differences more explicit in the last part of the introduction to the main text.

Q3: Fig. 3 presents the noise measured over a 10 MHz or 10 kHz bandwidth. It would be good to see a PSD of this noise so that we know what this bandwidth actually includes. According to Supplementary Table 2, the microwave mode linewidth is 13.7 MHz, so I'm not sure how these measurement bandwidths are chosen, and why the noise quanta measured with these two very different bandwidths can be directly compared, since neither is much larger than the linewidth.

A3: The PSD of microwave noise follows directly from the microwave linewidth (which also depends weakly on C), the mode temperature and the coupling ratios. We explicitly showed the PSD in our previous paper (Ref. 14) in the continuous wave domain and it is the same for this work which focuses on pulsed measurements.

The given unitless noise photon numbers always take into account the chosen resolution bandwidth (RBW), i.e. the detected power in photons/s in a 1 Hz RBW. Larger bandwidths are needed to resolve the full time dynamics accurately but those

measurements also capture more detection noise which reduces the SNR. So, the small RBW measurements are taken to resolve the smallest occupancies. The large RBW measurements are taken to measure the full time dependence. Together these results prove the point that the occupancy scales with the average applied power P_{avg} and not with the peak power P_{p} (relevant for C).

In the low RBW limit (compared to the emission bandwidth) we sample the maximum of the emitted noise peak. In the high RBW limit (compared to the emission bandwidth) we sample a larger part of the Lorentzian spectrum, which leads to slightly lower photon numbers when dividing the absolute detected power by the chosen resolution bandwidth. This mismatch between the emission bandwidth and the detection bandwidth leads to a nontrivial rescaling (on the order of up to a factor 0.8), which depends not only on the exact filter function, but also on C (because the microwave emission bandwidth changes slightly). Furthermore, to capture this in all its accuracy would also require taking into account small frequency shifts and changes of the emission bandwidth due to dynamical backaction. These details do not affect the main results and will be discussed in a future manuscript in detail.

Instead of adjusting both the measured experimental data and also the time dependent theory, and given the relatively small mismatch, we decided that it is most transparent to show what we actually detect and report the chosen RBW together with those numbers. The somewhat lower values at higher RBW are closer to a bandwidth matching scenario that is the most relevant one in practice, while the reported low RBW values represent a worst case scenario. It is important to realize that the lowest occupancies are taken from the low RBW data that capture the peak noise value. Those are the numbers reported in the abstract and conclusion and they remain unaffected by these subtle effects.

=> We added additional information about the required bandwidth mismatch to obtain high time resolution noise measurement data and the magnitude of the resulting impact on the backed out noise photon numbers with different RBW on page 4.

Q4: Can the authors say a bit more about what the significance or usefulness of the peak conversion efficiencies are? It doesn't seem like "pre-loading" the cavity with earlier photons should count toward the conversion efficiency of deterministically converting a particular signal photon. Unless there is a situation in which this is useful, I feel like the emphasis put on this in Fig 2c, d, g, and h might be misleading.

A4: The peak conversion efficiencies are achieved only for a small amount of time when the cavity is pre-loaded with photons. During this time, the loss due to finite coupling efficiency to the resonator does not limit the total conversion efficiency. In other words, the effective coupling ratio to the resonator is unity. As a result, we momentarily get a higher total conversion efficiency.

The higher conversion efficiency is relevant in situations when the signal photon is produced inside the transducer cavity, e.g. if a superconducting qubit is placed inside the 3D microwave cavity. This number also allows a fair comparison to other implementations where this is the case such as in the recent experiment by Caltech where the signal emitting qubit and mechanical mode are hybridized and no photon is produced and sent to the converter from outside.

=> We clarified this point in the main text of the manuscript.

=> We added a new Figure S7 to the Supplement to distinguish the effects of pre-loading from the overshoot due to the high cooperativity electro-optic interaction.

Q5: I found the discussion of Fig 4b and c somewhat confusing. There's a sentence that says "Use N_e and C , we can also infer the N_{out} that includes both thermal and quantum noise contributions..." Is this inferred N_{out} and the measured N_{out}^e the same, but just inferred for more values of N_e and C than what was measured?

A5: The calculated N_{out} for a given microwave occupancy N_e and C is shown in Fig. 3d and e as a density plot. The measured values of N_{out} are shown as color inside the circular symbols and agree very well. For Fig. 4 we then use the measured N_{out} to calculate N_e and mark the experimentally achieved values as crosses. The color scale is the calculated input noise N_{in} based on C and N_e (taken from measured quantities).

=> We changed the wording in this section to clarify the procedure. We also consistently use N_{out}^i with index i whenever we refer to both N_{out}^o and N_{out}^e together.

Q6: What is the meaning of the time delay between the peaks for different powers in Fig 3.b?

A6: The measured noise pulses are separated in time just for better visibility.

=> We have explicitly mentioned this in the figure caption now.

Q7: In the Supplement, the authors mention that the signal and LO are copropagating for the case of low cooperativity to maintain phase stability. Is this not a concern for higher cooperativities?

A7: Because of experimental limitations and convenience related to laser and optical filter locking, for higher pump powers, we were not able to implement co-propagating pump and signal optical paths. As a result, the phase stability timescale between these two paths was reduced from tens of seconds to tens of milliseconds. This is due to the instability of the overall phase in the two different optical fibers. Given the high bandwidth of our device we could still prove phase coherence by applying IQ modulated signals and faithfully reproduce the amplitude and phase after transduction, in particular also because the averaging times were significantly reduced for the high C and high efficiency transduction.

Thinking long-term this is not a limitation of the device but of the experimental setup which can be improved if the protocol requires it. On the timescale of ms it should be possible to implement either an analog lock or a digital post-processing correction.

=> We have clarified this point in the supplement.

Q8: I also could not find the meaning of the grey horizontal dashed line in Fig. 3.

A8: This line used to mark a value of 0.5 which is not particularly relevant for the output noise.

=> As such, we removed it.

Q9: What is P_p in the last paragraph of page 3?

A9: P_p represents the power during the time when the optical pulse is on.

=> This is defined in the main text now.

Q10: "grin" should be an acronym in the caption to Supplementary Figure 2.

A10: Implemented as suggested.

Reviewer 2

The manuscript presents the coherent transfer of weak coherent radiation through the electro-optical interface based on non-centrosymmetric crystals inside the dilution refrigerator that is promising for the direct interconnection of optical and microwave platforms. The result represents an original and important step in this direction at a nanosecond time scale and with a mean number of input quanta much less than unity. Optical and microwave heterodyne detection brings sufficient information to evaluate this milestone experiment regarding efficiency and added noise. The manuscript is well written, contains all relevant experimental details, and overcomes noisier, although more efficient, alternatives. As such, it is a timely, relevant and stimulating experimental result. It is worth being published in Nature Communication after solving these comments:

We thank the reviewer for the positive feedback. We have answered all the questions and comments below.

Q1: It will be better for a general reader to define the main text's visibly added noises N_{in} and N_{out} by formulas as they are central figures of merit.

A1: We agree that these terms should be better defined for the general reader.

=> We explicitly define the output noise photon number via the emission noise spectrum (defined in the SI) and the measurement filter function in the subsection "Thermal and quantum noise".

=> We explicitly defined the relationship between input and output noise spectra as well as the equivalent input noise photon number in the same subsection.

Q2: It will be stimulating for the community and also very good for the conclusion of the manuscript if critical predictions for the future transfer of at least nonclassical auto-correlation functions $g^{(2)}$ of single optical (microwave) photon to microwave (optical) output are estimated. The cross-correlation function after the conversion can also be evaluated for comparison. Also, will such an interface already allow transduction of Gaussian or single-photon entanglement?

A2: We agree that these are very interesting and important questions to address in the near future. A quantitative answer would require a detailed calculation of correlation

functions in the presence of finite sideband suppression (gain) and noise. This goes beyond the scope of this manuscript that focuses on understanding the time domain, and the first order effects such as the specific time domain response, the efficiency, noise, and gain for weak coherent signals. However, we point the reviewer to Ref. [4] (Zeuthen et al.) and in particular to Ref. 32 (Rueda et al.), where the transduction of non-Gaussian states and entanglement in electro-optic systems like ours is discussed in detail.

Generally speaking, with an equivalent input added noise of $\ll 1$ but an efficiency of < 0.5 from microwave to optics the device is suitable for most protocols that have been developed for lossy quantum optical networks. Specifically, it is ideally suited to perform entanglement protocols based on heralded single optical photon counting measurements as discussed in Ref. 33 (Krastanov et al.). We also believe that anti-bunching of microwave photon Fock states can successfully be demonstrated via optical photon counting (after transduction). Furthermore, the backed-out device properties are suitable to generate Gaussian electro-optic two-mode squeezing, i.e. deterministic entanglement generation in the continuous variable domain. Such experiments are currently in progress.

The device however is not yet suitable for fully deterministic and high fidelity conversion of non-classical states because the efficiency of < 0.5 results in added vacuum noise.

=> We have added further information in the introduction and also in the conclusion to better clarify the suitability of the device for certain protocols.

Q3: Based on the previous point, it will be good to specify then which regime of the device is optimal for particular tasks in 2).

A3: The lowest possible N_{in} should represent the ideal configuration for all quantum transduction scenarios that suffer from loss (non-unity efficiency). In our system this is achieved with intermediate C and pump power (lower C with lower noise is better for heralding schemes, intermediate to high C is important for deterministic schemes that also require near unity efficiency) and the lowest possible repetition rate to allow the device to fully cool down between the subsequent measurements (the technical limitation here is currently our way of locking the laser and filter cavities). This approach of low duty cycle does however limit the statistics and channel capacity because fewer experiments (transduction events) are performed in a given time.

=> We summarize these arguments in the conclusion.

Q4: Finally, it will be good to comment on a technical prospect of this device shortly. Can the noise be further lowered, and how? Can the total transduction efficiency be further increased, and how? Can we expect that the interface will improve in both parameters simultaneously soon?

A4: The easiest and most impactful improvement that can be made to the device is an improvement of the optical mode quality factor. At room temperature, we have produced resonators with 10-40 times higher quality factors but we have not managed to work with these at cryo temperatures yet. 10 times higher optical quality factors would reduce the optical pump power needed to achieve the same cooperativity by two orders of magnitude, thus significantly reducing thermal noise limitations.

In this paper, we have already achieved ~ 100% internal conversion efficiency (including gain). To achieve higher total conversion efficiency, we need better microwave and optical resonators (with high internal quality factors) such that they can be over-coupled and, thus, coupling losses can be minimized.

Additionally, the microwave-optical coupling 'g' can be increased by improving the transducer design - most importantly by reducing the mode volume. This is important to achieve a $C = 1$ before the Kerr effect kicks in. We are also working in this direction.

Finally, the use of other materials for the microwave cavity might allow us to thermalize better (faster) to the cold thermal bath of the dilution refrigerator.

=> We have added a paragraph discussing the limitations and the possible improvements of the presented transducer in the conclusion.

Referee 3

The manuscript by Sahu, et al., presents an electro-optic system for quantum conversion between GHz microwave signals and telecom optical photons. The converter system harnesses a 3D cavity design that couples a bulk lithium niobate (LiNbO₃) resonator with a superconducting aluminum cavity via the electro-optic effect. Although the converter is based on the author's previous design published in PRX Quantum 1, 020315 (2020), the authors have done substantial improvement here to push the device performance to close to the quantum-enabling regime. In particular, the authors implemented a pulsed optical pump scheme to boost the pump power, and hence the electro-optic cooperativity to near unity, without introducing a large amount of extra heat or thermal noise. As a result, a high conversion efficiency around 15% with impressively low subphoton added noise (referenced to the input).

The study of microwave-optical quantum conversion has recently become a pivotal research field that attracted a lot of attention. The realization of such quantum converters would enable powerful technologies, such as the combination of the state-of-the-art superconducting quantum computing units and long-distance optical quantum communication channels for building a future quantum network. This, nevertheless, is a very challenging task because of the demanding requirement of a high efficiency (at least 50%) with low added noise smaller than one for quantum signal conversion. So far, a quantum converter has not been demonstrated yet.

This work, in my opinion, represents a milestone effort toward the realization of microwave-optical quantum conversion. To my knowledge, this is the first demonstration of subphoton input added noise among all types of bidirectional microwave-optical converters. At the same time, a decent high conversion efficiency ~20% (although still below the 50% threshold) was achieved. These are very exciting and encouraging results that show the potential and advantages of the bulk electro-optic device design. Therefore, I strongly support its publication in Nature Communications.

The manuscript is well-written, and the data and results presented are solid and convincing. Below are some comments and questions that I think should be addressed before its publication.

We thank the reviewer for the detailed feedback and a positive outlook towards this work. We answer all the comments and questions below.

Q1: I feel it inaccurate to claim “Quantum-enabled operation” in the title. As far as I understand, even with zero added noise, a linear converter will need to have at least 50% efficiency in order to convert quantum signals (in other words, a non-zero quantum channel capacity). The efficiency and added noise achieved in this work are obviously below that threshold, which means the device here won’t be able to convert quantum information. Also, I don’t think I can consider any of the characterizations and measurements presented in the manuscript as quantum operations unless I missed something quantum-enabled.

A1: We picked the term “quantum-enabled” because $N_{in} < 1$ is what is needed to perform basic heralded quantum transduction, entanglement, and teleportation schemes and tried to distinguish it with this name from fully “quantum limited” operation, which would be needed to do deterministic conversion. Even though it is not as powerful as a fully quantum limited device, heralded quantum communication protocols will likely be with us for a while since any longer distance connection quickly suffers 50% of loss. In fact most quantum networking experiments are conducted in this limit today.

Finally, the device is also quantum-enabled in the sense that with these parameters we can deterministically produce entangled microwave-optical photon pairs e.g. in the continuous variable domain as explained in Ref. 32 (Rueda et al.).

=> We agree that our definition is not obvious and established in the field, so now we clarify what we mean with “quantum-enabled” in the first part of the introduction and also specify and distinguish it from a fully quantum-limited transducer that achieves $> 50\%$ efficiency and $N_{in} = 0$ as needed to beat the C. Caves limit for fully deterministic schemes that also preserve e.g. the negativity of the Wigner function of a non-classical state from input to output.

Q2: In the abstract, the authors claimed that “We achieve unity internal photon conversion efficiency by pushing the resonantly enhanced electro-optic interaction all the way to unity cooperativity”. A few comments here. (a) First, in the main text, it seems the highest cooperativity achieved is 0.92 – it is a bit rough to simply approximate 0.92 to one. Although extra data for $C = 1.21$ was shown at the end of the Supplementary Information, the system became unstable for conversion operation due to the optical parametric amplification. So, there is actually no data to support that the cooperativity can be continuously increased from 0.92 across 1 while maintaining the system in a stable condition. (b) I don’t see any definition or discussion about the “internal efficiency” through the manuscript. Although in a conventional EO converter the internal efficiency can be simply defined as $4C/(1+C)^2$, it is not straightforward from

Eq. (2) in this work since the an extra coupling mode is involved with cooperativity C_J and gain. How do the authors define “internal efficiency” in presence of the amplification from the Stokes mode? And will such definition still be useful if the “internal efficiency” may be larger than one? If authors don’t want to discuss the internal efficiency, “unity internal photon conversion efficiency” should not be stated in the abstract. Instead, highlighting the total efficiency could be more useful. (c) If I understand, the lowest input added noise of 0.16 was not obtained in the same condition as the highest conversion efficiency – the lowest noise was achieved at $C=0.2$, which gives a lower efficiency of 8%, while the highest efficiency (30%) was when $C=0.92$. So, it is a bit misleading in the abstract to emphasize the low added noise (“...with only 0.16 input noise quanta...”) then immediately state that a high “unity internal photon conversion efficiency” was also achieved. The author should indicate that the highest efficiency was obtained with a larger pump power.

A2:

(a) Cooperativity:

We performed measurements up to $C=1.2$ and showed these results in the SI together with theory. We could have included them in the main text but wanted to avoid distracting from the story by explaining the Kerr effect. The agreement at $C=0.99$ (yellow curves) is still very good since the Kerr effect kicks in in an exponential way above this value (red curves). We did not happen to have measured values at “exactly” $C=1$ and the measurement set with $C=0.99$ (shown in the SI) was obtained for a slightly different mode configuration as the one in the main text (which is why we did not include it in the other data set of the main text). Taking into account the experimental uncertainties, the difference between unity and 0.99 seems more of a technicality.

=> Nevertheless, we agree that values of C or bigger than C are not explicitly shown and we have revised the wording to “near-unity cooperativity” in the abstract.

(b) Internal efficiency:

We define the internal transduction efficiency as the total measured transmission coefficient divided by the (also experimentally characterized) external coupling ratios. This definition is independent of the transducer type, the physics of the device and does not rely on theory. In our view, such an operational definition that relies on measured values is important and more practical for the field rather than using specific theory to rescale the values somewhat artificially (taking into account finite transduction, gain and loss which can cancel each other). In the case of classical signal processing devices this would lead to a lot of confusion and we think the same might be the case of quantum signal processing devices.

To calculate the added noise correctly though, one needs to do the work nevertheless since any additional gain will lead to additional added noise. This is what we do in the current manuscript. We believe in the future one will (also operationally) make use of a quantum input signal such as a Fock state and measure what comes out. In that case unwanted gain would also naturally appear as added noise rather than gain + loss.

Having said all this, we understand that not only the reviewer but also the general reader wants to compare the obtained performance to the theory concept of pure conversion (without gain) and also to other transducers which do not (yet) suffer from gain.

=> We have therefore changed the wording to “internal transduction coefficient” to avoid confusion about how an efficiency can exceed 1. In addition, we now also discuss the origin of why it can be larger than 1 in the main text and added a new figure explaining the relevant effects in the SI.

=> In Fig. 2 we now also identify the pure (steady state) conversion efficiency as expected from an ideal theory model for each experimentally achieved cooperativity (blue circles) and report the highest achieved values (without gain) in the text.

=> In Figure 3 we highlighted the gain and made it clear that it is unwanted in the main text. Later on in the main text we identified that this gain is actually the limiting factor for significant added noise in the optics to microwave direction.

(c) Abstract:

There is no particular reason why the highest efficiency and lowest noise were not achieved at the same time - it is merely a result of the duty cycle that we chose as convenient in terms of the laser lock we had set up at the time.

=> In the abstract we now state the noise photon number and efficiency together for the same conditions.

The value of stating the internal efficiency is that it gives a sense of how close technical coupling improvements will bring the device to ideal operation. In our case the external in/out coupling losses are the only relevant limitations in terms of efficiency and we find this is important to point out.

=> We changed the wording “unity internal efficiency” to the actual obtained pure internal efficiency (without gain) of 95.5%.

Q3: It is amazing that the authors were able to send Watt-scale optical pump powers to the dilution fridge (even in the pulsed mode) while keeping the fridge at ~mK. The authors only provided the averaged pump power in the figures, but little information about how the pump affects the fridge and device was given. Although the average power could be as low as a few micro-Watt, there could be certain amount of light absorbed/scattered at the device interface, etc. Even 1% of one Watt (namely, 10mW) is significant for a dilution fridge because the typical cooling power at 20mK is only about 20uW. Can the authors comment on this? I hence believe that there must be non-negligible temperature increase of the base plate and the device at least for the high repetition rate measurement. Was there a thermometer directly mounted on the Al cavity for the authors to monitor its temperature? Or can the authors infer the temperature of the device based on the measured thermal noise N_e ? From Fig. 3(d,e), it seems the largest N_e is about 1.7 (I actually have a question about this, see next comment), corresponding to ~1K for a 8.8GHz mode, which is quite significant increase of temperature. I suggest the authors to explain these aspects and add relevant discussions in the main text, which will be useful to readers.

A3: Since both the microwave cavity and the dilution refrigerator (DR) have large thermal mass, nanosecond duration powers are not enough to cause any significant measurable temperature fluctuation using the standard resistive Ruthenium Oxide thermometer on the MXC plate. We find that it is the average power $P_{avg} = P_{peak} * pulse_length / trigger_time$ which affects the temperature of both the microwave cavity mode and the dilution refrigerator. The temperature dependence of the mixing chamber sensor in fact follows the one already reported in Ref. 14 (Hease et al.) Fig. 8 panel d - except, now it is the average power instead of the continuous wave power used in the previous paper.

Experimentally we made sure that the optical leakage during the pulse-off time is sufficiently low and we also designed the system such that the incident angle on the anti-reflection coated prism is perpendicular in order to outcouple as much light as possible - including reflected light. While we do not have a sensor directly mounted to the aluminum cavity, we can back out the electromagnetic microwave mode occupancy from the measured added noise at the device output (S_{out}) and, thus, its local temperature. This local temperature is understandably higher than the mixing chamber temperature of the dilution refrigerator because of the finite thermal conductivity of the superconducting aluminum cavity. It is also higher than the aluminum cavity walls

themselves, since the optical pulse heats up the dielectric inside, which also has finite thermal coupling to the microwave cavity and in turn to the dilution refrigerator bath.

In Ref. 14 we already discuss these effects in some detail and for example also show that part of the aluminum cavity undergoes the superconductor to normal metal transition for an effective microwave mode temperature of around 2 K (around 4 thermal noise quanta) for an average applied power close to 1 mW. For the highest power experiments reported in this manuscript with up to about 200 μ W of average power at most, we saw mixing chamber temperatures up to 80 mK (base temperature being 7 mK) in good agreement with Ref. 15 in Fig. 8d. For the demonstrated low noise transduction results we use slow repetition rates corresponding to average pump power of below 1 μ W with negligible increase of the dilution refrigerator base plate.

=> As requested we have added a paragraph discussing these effects at the of discussing Fig. 3 in the main text.

=> In the SI we added more relevant numbers and further information about the effective mode temperatures and the range of dilution refrigerator temperatures for the highest average pump powers of this pulsed experiment.

Q4: I am confused about the N_e label in Fig. 3(d,e) and Fig. 4(b,c). In Fig. 3(d,e), the N_e (circles in the plots) look much larger than the crosses in Fig. 4(b,c). But the figure captions said that they represent the same data in Fig. 3(c). So, was the x-axis label in Fig. 3(d,e) wrong? Another confusing part is that it is not clear which points in Fig. 3(c) correspond to the four circles in Fig. 3(d) since there are 8 points (four blue four red) in Fig. 3(c) for $C=0.38$. Also, in Fig. 4(b,c), do the five crosses correspond to the first five blue dots in Fig. 3(c) (there are seven blue dots in total)?

A4: The N_e values in Fig. 3 and Fig. 4 are inferred from the raw noise data reported in Fig. 3 a-c. In case of Fig. 3 d and e we show the fast time domain data obtained with high filter bandwidth and fast repetition rates, while for Fig. 4 we focus on the lowest occupancy data sets, which have been measured with a smaller filter bandwidth (blue points in Fig. 3c).

=> We have made the wording more explicit in the captions so the reader knows exactly which raw data points of N_{out} have been used to obtain the shown values of N_e .

Q5: Although the device details are shown in the Methods, it would be convenient to give one- or two-sentence description of the device in the main text for readers who are not familiar with the authors' previous work.

A5: => As suggested, we have added additional information about the device design to the main part of the manuscript.

Q6: In Fig. 2, the color contrast for the measurement and theory lines are a little too low to be seen especially for the light/dark orange lines. Another thing is that the caption says "bright dots are calibrated measurements" – I actually couldn't find any "bright dots" at the beginning, then I realized that the dots are so dense that they form a continuous thick line with light color. Please somehow make them clearer.

A6: => We increased the contrast and included only independent points at the experimental 200 MHz resolution bandwidth (5 ns time resolution) in the plot now (compared to the oversampled data with 1 ns shown before).

Q7: In Fig. 2(d,h), could the authors explain why the dashed orange lines can be larger than one for the internal efficiency? Is it because of the preloaded cavity photons? However, it barely makes sense to me to define an internal efficiency that can be larger than one. The author may want to re-define it and scale it down to within one so that it can be used to compare with other converters.

A7: In part this question has been addressed already in our answer A2(b). In short,

=> we introduced the pure (steady state) conversion efficiency now that is bound to below or equal 1 as expected and as shown in Fig. 2 now (blue circles).

=> we further clarified that the internal transduction coefficient (steady state) can be higher than 1 because of transducer gain (blue squares vs. blue circles).

=> We added a discussion to the main text and a new section 5: "Time dependence of transduction efficiency" including a new Fig. 7 to the SI that explains the reasons for significantly higher peak transmission coefficients (red symbols), which is indeed mainly due to pre-loaded cavity photons but also related to gain and electro-optic oscillations that appear for cooperativities close to 1 and above.

Q8: On Page 3, bottom left, it would be useful to give the value of the absolute pump power used for 3.16×10^{10} pump photons. I guess it would be around one Watt?

A8: It is 0.4 W and we have now added this information.

Q9: I saw that different suppression ratio S was used throughout the experiments. I am wondering how the authors control or tune S ? I thought it is determined by the cross-polarization coupling rate J , which is fixed after the lithium niobate resonator is fabricated.

A9: We used a different set of modes to change the suppression ratio S for one particular measurement in order to highlight the effect of vacuum amplification (in Fig. 2 a lowest curve). As we go over different FSRs of the optical resonator (different absolute optical wavelength), we find numerous modes and mode families with different magnitudes of avoided crossings with a choice to pump them on either side and, thus, realize different suppression ratios for the conversion process.

=> We have added a sentence to clarify this in the manuscript.

Q10: In the conclusion paragraph, I am not sure that the authors can state the input added noise is “much smaller” than 0.5 ($0.16 \ll 0.5$?). It might be okay to write “ $N^{\text{oe}} < 0.5$ ” or “ $N^{\text{oe}} \ll 1$ ”.

A10: We agree, and changed the wording correspondingly.

Q11: In the conclusion paragraph, can the total efficiency of 15% and lowest input noise N^{oe} be obtained at same condition (namely, with same C and S)? I thought that the 15% efficiency is when $C=0.49$, and the 0.16 added noise is when $C=0.2$ which only gives 8% efficiency. Please clarify this to avoid any misunderstanding.

A11: We state the input noise photons and efficiency together (for the same conditions) in the abstract now and have removed this information from the conclusion in order to avoid repetitions.

Q12: As I understand, the microwave cavity is made from Al, which should be superconducting below 1.2K. However, the intrinsic microwave Q factor seems very low,

only ~1000. So, is the Al cavity superconducting during the experiments? If yes, then what are the major microwave loss sources? From the bulk lithium niobate?

A12: For the experiments presented here the Al cavity is superconducting (please also refer to our A3). The superconductor to normal metal transition happens at around 1 mW average optical power as shown in our previous manuscript (Ref. 14).

The origin of the microwave losses is a matter of active investigation and this is our current verdict: the smallest part is due to seam losses in the Al cavity, a larger part due to intrinsic losses of the lithium niobate and likely the dominant contribution we attribute to piezo-electric losses that are enhanced due to the tight clamping along the WGMR rim. We had already discussed this issue briefly in our previous manuscript (Ref. 32) and suggested a center clamped design. Initial tests with such designs and also using different materials confirm that the microwave Q can be improved by factors of around 3-5 so far. More experimental and numerical simulation work is needed to arrive at a quantitative conclusion in this regard. Earlier results show microwave Qs of $>10^4$ for bulk crystals that were very carefully suspended (<https://doi.org/10.1103/PhysRevB.92.060406>). Here the limit was identified to be frequency and material dependent natural impurities.

=> We have added a summary of this point to the new concluding paragraph that provides an overview of the limitations and an outlook. We also included the mentioned references.

Q13: At the end of the main text, could the authors provide some discussions regarding how to further improve the device performance to achieve a real quantum converter? For example, there seems to have quite some room to improve the microwave Q, which will help boost the cooperativity and also lower the pump power needed. I think such discussions could benefit the researchers in community to move forward.

A13: Please refer to the answer just above as well as the answer to Q4 from Referee 2.

=> We have added a discussion of realistic and required improvements in the conclusion of the paper.

Q14: In the SI experimental setup figures, could the authors provide more information about the filters F1, F2? What are the model and specifications (bandwidth, suppression ratio, etc)?

A14: As requested we have added the missing information about the filter properties in the corresponding SI figure caption.

Q15: In the SI Note 7 Kerr Effect, the authors state “For longer optical pump pulses, the cooperativity threshold for amplification becomes smaller and well below unity.” Could the authors explain more about this? Does this mean for longer pump pulses, $C=1$ can no longer be achieved?

A15: Yes, this is correct. We have observed that the power threshold for Kerr-based instability drops as the pulse length is increased. The unstable amplification occurs at the very end of the pulse (refer to SI figure) and can be avoided by just making the pulse shorter. At this point, the power in the pulse can be increased until amplification appears again at the end of the pulse.

For longer pump pulses, $C=1$ can be achieved if the $\chi(2)$ mediated microwave-optical coupling ‘g’ is improved, or if the microwave quality factor is improved. We believe that improving the optical quality factor reduces the power threshold for both $\chi(2)$ and $\chi(3)$ processes simultaneously and thus, has no advantage in avoiding the $\chi(3)$ process at higher C.

=> We have made this point clearer in the main text and the SI. Also, in the outlook we now emphasize that it might be important to increase the electro-optic coupling to avoid being limited by the Kerr nonlinearity at high pump powers.

Reviewers' Comments:

Reviewer #1:

Remarks to the Author:

The authors have addressed most of my comments from the previous round of reviews. However, I am still a bit concerned that the abstract and conclusions are misleading in how it quotes the values for added noise and conversion efficiency, as also pointed out by referee 3. In abstract of the revised version, there is one sentence stating the added noise and total efficiency for $C = 0.2$ and $C = 0.49$ as far as I can tell, but those cooperativities are not stated. The next sentence then talks about near-unity cooperativity ($C = 0.92$) and internal conversion efficiency, which can easily mislead readers into thinking that those two sentences refer to the same conditions, which they do not. Similarly, in the conclusions, there's a sentence that talks about a total efficiency of 15%, a $N_{in} \ll 1$, and ends with "for a cooperativity of $C = 0.2$ ". The caption to figure 3 states that for $C = 0.2$ the efficiency is 8%, not 15%. Also, the abstract states that for 15% efficiency the added noise is 0.41, which I would say is not $\ll 1$. So, I think that it's very confusing to quote all these numbers in the same sentence unless it's clearly stated that they correspond to different circumstances.

I think it's important that the authors address this issue. Otherwise I support publication.

Reviewer #2:

Remarks to the Author:

The authors answered all the questions and points I raised in the report and made adequate changes to the revised manuscript. I suggest this manuscript be published in Nature Communication.

Reviewer #3:

Remarks to the Author:

The authors have addressed all my comments and improved the manuscript accordingly. I believe that the revised manuscript is now suitable for publication in Nature Communications.

We thank all three referees for their positive feedback. Below we address the remaining concerns of referee 1.

Reviewer 1:

The authors have addressed most of my comments from the previous round of reviews. However, I am still a bit concerned that the abstract and conclusions are misleading in how it quotes the values for added noise and conversion efficiency, as also pointed out by referee 3. In abstract of the revised version, there is one sentence stating the added noise and total efficiency for $C = 0.2$ and $C = 0.49$ as far as I can tell, but those cooperativities are not stated. The next sentence then talks about near-unity cooperativity ($C = 0.92$) and internal conversion efficiency, which can easily mislead readers into thinking that those two sentences refer to the same conditions, which they do not.

=> As required in the last round, we now state the relevant pairs of numbers explicitly, i.e. those that are valid for the same conditions and cooperativity. We are happy to add the C numbers for each case if the referee insists but personally we feel that it would make the abstract even more technical and do not see the added value since it is clear from the different efficiency that it is for two different C s. Also, we were asked to shorten the abstract.

=> We do agree however that it can lead to confusion in combination with the next sentence where we mention that near-unity cooperativity is accessible. In order to fix this, we added the word "also". While this is a small change it makes it clear that it is a different condition compared to the two mentioned before.

=> As requested by the editor we also shortened the abstract to meet the 150 word limit by removing the last sentence and the part "In analogy to a fiber optic modem,".

Similarly, in the conclusions, there's a sentence that talks about a total efficiency of 15%, a $N_{in} \ll 1$, and ends with "for a cooperativity of $C = 0.2$ ".

=> We agree and have updated the value of C to match with 15% and replaced the ' \ll ' symbol with ' $<$ ' so the sentence is now for the same experimental condition and correct.

The caption to figure 3 states that for $C = 0.2$ the efficiency is 8%, not 15%. Also, the abstract states that for 15% efficiency the added noise is 0.41, which I would say is not $\ll 1$. So, I think that it's very confusing to quote all these numbers in the same sentence unless it's clearly stated that they correspond to different circumstances.

=> We have addressed these concerns with the above mentioned changes.

I think it's important that the authors address this issue. Otherwise I support publication.